# The Clinical Relevance of the Microbiome in Hidradenitis Suppurativa: A Systematic Review

**DOI:** 10.3390/vaccines9101076

**Published:** 2021-09-25

**Authors:** Dillon Mintoff, Isabella Borg, Nikolai Paul Pace

**Affiliations:** 1Department of Pathology, Faculty of Medicine and Surgery, University of Malta, MSD 2080 Msida, Malta; isabella.borg@um.edu.mt; 2Department of Dermatology, Mater Dei Hospital, Triq Id-Donaturi tad-Demm, MSD 2090 Msida, Malta; 3Centre for Molecular Medicine and Biobanking, Faculty of Medicine and Surgery, University of Malta, MSD 2080 Msida, Malta; nikolai.p.pace@um.edu.mt; 4Medical Genetics Unit, Department of Pathology, Mater Dei Hospital, MSD 2090 Msida, Malta

**Keywords:** hidradenitis suppurativa, microbiome

## Abstract

Hidradenitis suppurativa is a chronic disease of the pilosebaceous unit. The name of the condition is a testament to the presumed relationship between the disease and the microbiome. The pathophysiology of hidradenitis suppurativa is, however, complex and believed to be the product of a multifactorial interplay between the interfollicular epithelium, pilosebaceous unit, microbiome, as well as genetic and environmental factors. In this review we assimilate the existing literature regarding the role played by the human microbiome in HS in various contexts of the disease, including the pathophysiologic, therapeutic, and potentially, diagnostic as well prognostic. In conclusion, the role played by the microbiome in HS is extensive and relevant and can have bench-to-bedside applications.

## 1. Introduction

Hidradenitis Suppurativa (HS) is a complex condition of the pilosebaceous unit (PSU). Patients experience multiple nodules, abscesses, and draining fistulae in intertriginous skin, which cause a significant impact on their quality of life [1]. The pathophysiology of HS is multifaceted and is the product of interactions between inflammation, a genetic tendency, the microbiome, and environmental factors [2]. The name Hidradenitis Suppurativa is misleading as it suggests the condition to be the result of purely infective processes involving the apocrine glands. This name is the subject of controversy, with some recommending that the disease should be renamed Acne Inversa [3,4]. The latter name conceptualizes the fact that unlike acne, patients with HS experience deep seated abscesses which involve skin that is not usually affected by acne (intertriginous skin), and which therefore hosts a distinct microbiome [5]. The propensity of HS to manifest as a purulent disease involving apocrine gland-bearing skin is undeniable. However, dismissing the pathophysiological processes at play as purely infective would be an oversimplification.

The intricate role of the microbiome has been investigated in various cutaneous disorders, including psoriasis, atopic dermatitis, and other diseases of the PSU such as acne [6,7,8]. In this review, we scrutinize the existing literature with regard to how the skin microbiome in HS skin differs from that in healthy skin, whether it incites or potentiates disease, how it alters in response to therapy, and whether it can be manipulated to alter disease trajectory.

## 2. Materials and Methods

A review of the literature pertaining to hidradenitis suppurativa and the cutaneous microbiome was conducted in PubMed, Science Direct, and Google scholar. The following Medical Subject Heading (MeSH) terms were used: “hidradenitis”, “suppurativa”, “acne”, “inversa”, “pathophysiology”, “microbiome”, “cutaneous”, “gut”, “intestinal”, “dysbiosis”, “biofilm”, “antibiotic”, “bacteria”, “fungi”, “probiotic”, “biomarker”. The Boolean operators used were “AND” and “OR”. Genomic studies investigating changes in the composition or characteristics of the skin microbiome in human or animal studies were included. Additionally, studies exploring the role of the gut–skin axis in HS, and the effects of diet or antibiotics in modulating HS skin microbiomes and disease risk, progression, and outcome were also included. The search covered articles published between January 1990 and July 2021 and was restricted to articles published in the English language. Furthermore, handsearching and citation review of relevant studies were also conducted to identify studies that were not captured by electronic database search.

### Exclusion Criteria

Studies were excluded if they did not directly investigate the association between the cutaneous microbiome and HS. More specifically, articles whose primary objective was the investigation of skin composition or gut microbiome in metabolic syndrome or its components were not included. Articles not written in the English language, reviews, editorial letters, and comments were excluded.

## 3. Results

### 3.1. The Human Microbiota

The human microbiota describes a complex set of microorganisms that colonise the skin and several mucosal environments, including the gastrointestinal, respiratory, and urogenital tracts. These microorganisms form an elaborate and discrete ecosystem that maintains a commensal relationship with normal host physiology. Additionally, changes in the microbiota are intricately implicated in several disease processes through the modulation of metabolism and immunity. The magnitude and diversity of the human microbiota relative to somatic cells is extensive. Recent estimates suggest a ratio of 1.3 bacterial cells per human cell, and an estimated 500–1000 bacterial species inhabit the human body at any one time [9,10]. These estimates do not consider fungi and viruses. Furthermore, microorganisms exhibit significant diversity, each having several genetically unique subspecies. Collectively, the microbial genome (microbiome) catalogue exceeds the estimated 20,000 human genes by several orders of magnitude [11,12]. The human microbiome is not static but evolves in response to a wide array of host elements, including age, lifestyle, diet, endocrine factors, and underlying disease. In addition, a significant extent of interindividual heterogeneity is reported even in the absence of disease [13]. These factors showcase the dynamic nature of human microbiome ecosystems which have coevolved to adapt to constantly changing host physiological conditions. They also complicate our understanding of changes in the microbiome that are causally associated with disease or reflect ongoing pathology. Alterations in the composition of human microbiota (or dysbiosis) has been associated with several diseases ranging from inflammatory bowel disease, malignancies, asthma, type 1 and type 2 diabetes, to major depressive disorder [14,15,16,17,18,19]. 

### 3.2. Surveying the Skin Microbiome 

The past decade has witnessed a surge in microbiome research, driven by methodological advances in the field. The two principal sequencing-based approaches for microbiome research are metataxonomics and metagenomics [20]. Metataxonomics involves the targeted amplification and sequencing of marker genes that are specifically conserved across taxa, such as the 16S ribosomal RNA (rRNA) gene in bacteria and the internal transcribed spacer 1 (*ITS1*) in fungi. These genes are characterized by hypervariable regions that enable species identification, and contain conserved regions that can be targeted by universal primers [21]. Metataxonomic techniques enable the fast and cost-effective identification of microorganisms. However, they possess several drawbacks. They cannot capture viral sequences, and short-read sequencing of amplicons cannot offer species-level resolution. Additionally, some studies have highlighted the limitations of sequencing 16S rRNA for bacterial detection [22,23].

Metagenomics is based on the random shotgun sequencing of microbial DNA without the prior selection of specific loci. This technique enables a de novo assembly of genomes and is free from amplification bias as it sequences all genetic material in a given sample. Crucially, metagenomic approaches allow for the assessment of the relative abundances of different kingdoms and enable a high-resolution assessment of microorganisms at the species and strain levels [24]. These constitute significant advantages over amplicon-based approaches and explain the increasing popularity of metagenomics-based analyses. Metagenomic studies require a high read count, require alignment to a reference genome for classification, and can capture a large quantity of reads from the host organism [25].

Complementary multi-omics approaches are also available. These techniques supplement sequencing data with transcriptomic, proteomic, and metabolomic data for the improved profiling of microbial communities [26]. A detailed overview of microbial multi-omics approaches is provided by Franzosa et al. [27]. Although beyond the scope of this article, such approaches provide direct insight into the functional activity of the microbiota at the transcriptional and metabolic levels, as evidenced by proteomic and metabolomic studies in the human gut [28,29]. As techniques for the integration of multi-dimensional datasets develop further, an improved understanding of microbiota biology in health and disease is anticipated, fuelling translational advances in biomarker discovery or treatment development.

Human skin provides a unique environment with low microbial biomass, a high contamination risk, and a wide diversity of microbiota, coupled with site-specific microorganisms. These elements necessitate important considerations for microbial studies to avoid bias. It is essential for both researchers and clinicians to understand that the comparability of microbiome research can be limited by several elements. These include sampling and methodological variation between studies, and laboratory factors during DNA extraction and downstream processing [30,31,32,33,34]. Kong et al. reviewed skin microbiome research standards and highlighted several factors and pitfalls to consider when performing or interpreting skin microbiome research [35]. 

### 3.3. Composition and Diversity of the Human Skin Microbiome 

As the body’s largest interface with the external environment, the skin is home to an extensive array of symbiotic and commensal microorganisms. The skin microbiota orchestrates several key homeostatic processes, including protection against pathogens, immune development, and the breakdown of natural products through a bidirectional metabolic exchange with host cells [36,37]. Disruption of the natural balance between commensal and pathogenic microbes can lead to skin or even systemic disease, and thus a detailed understanding of the properties of healthy skin microbiome is presented next.

The composition and relative abundance of skin microbial taxa hinge primarily on the physiological attributes of the skin microenvironment, with notable changes between moist, dry, and sebaceous sites. Distinct skin niches that differ in moisture, sebum content, topography, pH, UV exposure, and temperature are colonised by different bacterial taxa in a site-specific manner [38]. Metagenomic sequencing studies in healthy adults demonstrate that sebaceous regions are dominated by members of the lipophilic *Cutibacterium* (formerly known as *Propionibacterium*) and *Staphylococcus* genera. Moist areas such as the antecubital and popliteal fossae are favoured by members of the *Corynebacterium* and *Staphylococcus* genera [39,40]. Relatively dry regions exposed to fluctuations in temperature such as the limb surfaces exhibit less microbial diversity. These sites are primarily dominated by *Proteobacteria*, *Bacteroidetes*, and *Actinobacteria*. Conversely, the fungal skin composition is relatively uniform across core body sites. Members of the genus *Malassezia* are found throughout the body, although a higher fungal diversity is reported in the feet, with combinations of *Aspergillus, Cryptococcus,*, *Rhodotorula*, *Epicoccum,* and other species [41]. Across all body sites, bacteria are more abundant than fungi. While several studies on the cutaneous microbiome have focused on bacteria and fungi, the diversity of skin viral communities are less well understood. Eukaryotic DNA viruses are specific to individuals rather than anatomic site, with no core DNA viral sequence being conserved across different individuals [42]. Despite the broad variation in microbiome composition and structure across different body sites, and the extensive exposure to the external environment, longitudinal sampling studies have shown that in the healthy adult, microbial communities remain largely stable over time [43]. Bacterial and fungal species at sebaceous sites are the most stable, whereas microbial species in the feet are the least stable. The long-term stability of the human skin microbiome thus bears similarity with the longitudinal maintenance of the human gut microbiome [44].

Skin metagenomic studies also enable the high-resolution examination of skin bacterial communities at the level of strain diversity. Different bacterial strains can exhibit variation in transmissibility, virulence, antibiotic resistance, and metabolism. Consequently, strain resolution analysis can have implications for health and disease. Oh et al. applied strain analysis to the common skin commensals *Cutibacterium acnes* and *Staphylococcus epidermidis.* They showed that differences in the *S. epidermidis* strain distribution were primarily driven by anatomic site, whereas the strain distribution of *Cutibacterium acnes* is individual-specific rather than site-specific [39].

The colonisation of specific skin microenvironments by different taxa hinges on the capacity of resident microbiota to utilise resources in the skin. Compared to the human gut, the skin is deficient in carbohydrates and rich in lipids, and thus demonstrates a relative paucity of nutrients capable of supporting microbial growth. Skin appendages such as sweat glands, sebaceous glands, and hair follicles can create conditions favouring or inhibiting the growth of specific microorganisms. For example, *Cutibacterium acnes* is a facultative anaerobe that uses proteases to liberate arginine from skin proteins and lipases to degrade sebum triglycerides into free fatty acids [37,45]. The free fatty acids produced by sebum lipid hydrolysis inhibit the growth of Gram-positive pathogenic bacteria such as *Staphylococcus aureus* and *Streptococcus pyogenes* [46]. Sweat contains antimicrobial peptides that inhibit microbe colonisation, and its evaporation acidifies the skin environment, rendering it unfavourable for the growth of specific organisms [47].

### 3.4. Skin Microbiota–Immune System Crosstalk 

The immune system is central to the maintenance of commensal skin microorganisms and the elimination of pathogenic microbes. Extensive interplay occurs between the innate and adaptive arms of the immune system, keratinocytes, and skin microbiota. Keratinocytes express pathogen recognition receptors that recognize and bind to pathogen-associated molecular patterns, triggering microbicidal and pro-inflammatory innate immune responses to eliminate or contain harmful pathogens [48]. Skin epithelial cells produce a wide array of anti-microbial peptides (AMPs), including several proteins (defensins, cathelicidins, and others) that possess anti-bacterial, anti-fungal, and anti-parasitic activity [49,50]. Skin commensals function in the development and regulation of cutaneous adaptive immunity [51]. The neonatal period constitutes a crucial window for establishing immune tolerance to commensal microbes. The host–commensal symbiosis in human skin is dependent on regulatory T cells [52]. Following the initial period of immune tolerance, skin commensals elicit distinct effects on the immune system through interleukin-1 signalling, where the induction of effector T cells occurs in the absence of classical inflammation [53]. The skin microbiome thus plays a critical role in the induction and education of host immunity, enabling sustained tolerance to innocuous antigens, a process termed “homeostatic immunity” [54]. Conversely, when skin commensals are introduced systemically such as in the setting of a skin barrier breach, a classical inflammatory response is elicited. This showcases how the microbiome–host relationship is context-specific, ranging from commensal to pathogenic and depending on host factors, and how the dysregulation of the host–microbiome dialog can result in chronic inflammatory host responses.

### 3.5. Cutaneous Dysbiosis 

The skin microbiota is a necessary component of the immune barrier, with microorganisms preventing the colonisation and growth of pathogenic bacteria—a process known as colonisation resistance. Its transition to an altered state, or dysbiosis, involves a change in the structural and functional balance of the microbiome, and is associated with both systemic and localised dermatological disorders. In the setting of cutaneous dysbiosis, the pathogenic potential of the microbial components of healthy skin can be activated. Several common skin pathologies such as acne, atopic dermatitis, and chronic wounds are correlated with an underlying microbial component. In acne, the inflammation of the pilosebaceous unit is associated with the proliferation of lipophilic bacteria, specifically *Cutibacterium acnes*. This bacterium secretes hyaluronases and lipases that damage the pilosebaceous unit and encodes immunogenic proteins that activate classical and alternative complement pathways [55,56]. Multiple studies have demonstrated that the presence of *Cutibacterium acnes* biofilms in follicles and the type 1A1 phylogroup are associated with acne development [57,58]. In atopic dermatitis, the relative abundance of *Staphylococcus aureus* and *Staphylococcus epidermidis* increases during flare ups, although the functional role of staphylococci in driving disease is not fully understood [59,60]. Microbial changes with reduced diversity in unaffected skin of adults prone to atopic dermatitis have been described [61,62]. Microbiome research in both acne and atopic dermatitis have provided valuable insight into how commensal bacteria can initiate or exacerbate skin disorders, often against a background of skin barrier defects and/or altered immunity. Dysbiosis is common to many skin diseases, although distinguishing whether dysbiosis is a cause or effect of disease is a challenging question to address [7]. In the next section, a systematic overview of cutaneous dysbiosis in HS is presented.

### 3.6. Cutaneous Dysbiosis in Hidradenitis Suppurativa

Several investigations have characterized the skewed cutaneous microbiome that distinguish HS. Next generation sequencing (NGS) targeting 16S rRNA has been employed in different contexts of the disease. A summary of key studies employing this technique is provided in Table 1.

HS skin appears to have a microbiome distinct from that of healthy skin and which is overwhelmingly populated by anaerobic Gram-negative bacteria [68]. Differences in HS microbiome can be detected even in the absence of clinically observable lesions [69]. Differential abundance gene testing (which refers to the statistical method of investigating the significance of specific bacterial species abundance in two different ecosystems) showed significant differences between lesional and healthy (control) skin as well as between lesional and non-lesional skin [70]. The comparison of non-lesional and healthy skin, however, shows no significant difference in bacterial species abundance [71]. Functional analysis also confirms significant metagenomic differences between lesional and healthy control skin as well as between lesional and non-lesional skin, but not between non-lesional and healthy control skin [71]. The microbiome can vary even between HS-lesion type, with *Staphylococcuslugdunesis* being associated with nodules and abscesses, polymicrobial anaerobes with chronic suppurating lesions, and anaerobic *Actinomycetes* with severe suppurating lesions [72].

Firmicutes constitute the broadest bacterial phylum in the skin of both healthy individuals as well as HS patients, with the genera *Staphylococcus* and *Streptococcus* predominating. *S aureus* and coagulase negative staphylococci (CNS) have been identified as the most common bacterial species at various skin depths in HS [73]. Conversely, none of the 27 HS patients recruited in a retrospective study investigating the microbiome in resection specimens had *Staphylococcus* spp. identified by IF/FISH, peptide nucleic acid-fluorescence in situ hybridization (PNA-FISH), and immunostaining with anti-Gram-positive antibodies [74]. Nasal colonisation of *S. aureus* has been found to be significantly reduced in patients with HS when compared to healthy controls [75]. Notwithstanding, and despite having higher levels of the antimicrobial chemokine Il-26 in plasma, HS patients appeared to be more susceptible to staphylococcal infections [76]. This may be explained by the fact that unlike healthy controls, neither Il-26 nor toxic shock syndrome toxin 1(TSST1)-stimulated peripheral blood mononuclear cells (PBMC) inhibit *Staphylococcus aureus* growth. In these conditions, HS-PBMC cytotoxicity is reduced when compared to healthy controls [76]. Additionally, elevated levels of Il-26 have been associated with sustained local inflammation [77]. 

Other deficiencies in HS skin immune response to pathogenic organisms include decreased concentrations of the antimicrobial peptide Dermcidin, which is naturally found in human sweat [78,79]. Dermcidin is also found in lower concentrations in plasma as well as in the lesional skin of acne patients when compared to controls, and increases upon treatment with low or conventionally dosed isotretinoin, a retinoid which has been used with varied success for the treatment of HS [80,81,82,83,84].

Another emerging pathogenic factor in various skin disorders, including HS, is the biofilm [85]. Biofilms represent microbial aggregates suspended in a self-produced slurry of extracellular polymeric substances [86]. A histological evaluation of HS skin reveals that it is inundated with biofilm [87]. Ten HS patients whose skin was affected with inflamed nodules revealed biofilms in two of the patients, suggesting that biofilm-associated bacteria are not only related to chronic lesions such as sinus tracts, but also acute lesions such as nodules [88]. Unaffected HS axillary skin also demonstrates the presence of biofilm, supporting the hypothesis that HS is a disease of bacterial biofilm [89]. 

Certain processes favour the tendency of HS skin to host a biofilm. *S. epidermidis*, a usually non-virulent commensal in healthy skin, has been shown to have a propensity for planktonic growth and biofilm production in both lesional and non-lesional HS skin [90]. This bacterium is the abundant CNS in HS patients [91]. A matter of concern is *S. epidermis* in lesional skin, which demonstrates resistance to guideline-recommended antibiotics such as tetracyclines and clindamycin [90,92]. The biofilm-producing potential of other CNS such as *Staphylococcus lugdunensis* is also enhanced in HS-lesional skin when compared to non-lesional, control, and reference strains [93].

HS patients with identified biofilms are found to have an increased concentration of CD4+ cells when compared to healthy controls (with or without biofilm) and HS patients without biofilm. These findings suggest that biofilms may stimulate the production of regulatory T cells, resulting in an intensified immunomodulatory response [89]. On the other hand, healthy patients have a tendency to host biofilms, which is cocci-predominant [94]. This suggests that the composition of the biofilm partly dictates cutaneous homeostasis. 

As a disease of the PSU, the next step in decoding the complexity of HS is to investigate the composition of the follicular microbiome. Metataxonomic studies targeting bacterial 16S rRNA and eukaryotic 18S rRNA genes performed on cutaneous biopsies revealed follicular lesional skin to contain an abundance of *Corynebacterium*, *Porphyromonas,* and *Peptoniphilus* species, whilst *Acinetobacter* and *Moraxella* predominated non-lesional skin [64]. As a species, *C. acnes* and *C. striatum* are significantly more abundant in follicles of healthy controls when compared to lesional skin [64]. The richness of bacterial species does not appear to differ between HS and healthy controls [64]. 

The role played by individual bacterial species in HS pathology has been reviewed [95]. However, it is clinically relevant to recognize that some organisms, such as *Actinomyces schaalii*, may exacerbate the disease state by inducing abscess formation, which may be unrelated to HS and the PSU altogether [96]. Nodules secondary to deep dermatophytosis in patients with HS can also be mistaken for the malignant transformation of chronically affected HS skin, which is a known association [97,98,99]. The rapid identification of such pathogens, including slow-growing organisms such as *Actinomyces neuii* is essential in avoiding confounding symptomatology [100].

### 3.7. Beyond the Skin Microbiome 

Although HS manifests in the skin, other microbiomes such as the gut microbiome have the potential of mediating disease by facilitating systemic inflammatory pathways supported by the gut–skin axis [101]. This pathophysiological axis has been characterized in atopic dermatitis, psoriasis, and rosacea [102,103,104]. Enteric microbiota can influence the skin microbiota, resulting in cutaneous dyshomeostasis, which can ultimately lead to skin pathology, including that of the PSU such as acne [101]. 

Congruent with other pathophysiological theories, gut dysbiosis may also potentiate HS by mediating inflammation. It has been hypothesised that enteric dysbiosis (potentially caused by a high-fat diet) elevates inflammatory cytokines such as Il-1β, Il-17, and TNF-α. This ultimately leads to an increased level of matrix metalloproteinases and subsequently HS lesion formation in susceptible individuals [105]. 

The 16S rRNA amplification techniques of faecal samples have been employed to compare enteric and cutaneous microbiome of HS patients and healthy control subjects [106,107]. Worthy of note is *Robinsoniella,* which was exclusively detected in the faces of HS patients [106], a feature HS shares with rheumatoid arthritis and ankylosing spondylitis [106]. It appears that enteric species diversity may be reduced in patients with HS in keeping with other inflammatory conditions such as inflammatory bowel disease (IBD) [108]. 

Indirect evidence of the specific role played by the microbiome can be derived from co-morbid disease. For example *Faecalibacterium prausnitzii* abundance was depleted in patients with concomitant HS and IBD, but not in patients suffering from HS exclusively [109]. The relative abundance of *F. prausnitzii* does not appear to be diminished when compared to healthy controls. Subspecies enrichment of *F. prausnitzii* has been associated with other inflammatory dermatoses, including atopic dermatitis [110]. 

The gut microbiome may also unravel molecular processes linking HS with obesity, a significant risk factor for the disease [111]. In murine models, a positive correlation between the abundance of *Firmicutes* in the gut microbiome and body weight has been reported, while *Bacteroides* spp. are negatively correlated with body weight. *Lactobacillus* relative abundance is also increased in mice gaining weight from a high fat diet [112]. Established HS risk factors such as obesity may then exert a direct influence on the cutaneous microbiome, forming a feedback loop. It has been shown that the *S. aureus*, *Enterococcus,* and *Proteus* are more likely to be cultured in obese HS than lean HS patients [113]. Visceral obesity is accompanied by a systemic, subclinical, proinflammatory state. This is partly mediated by IL-16, IL-1, and TNF-α. There are also important differences in the gut microbiome associated with obesity and the metabolic syndrome, which should be considered as confounders in metagenomic HS studies [114]. 

The microbiome extends beyond skin and gut. Bacterial translocation in peripheral blood is an established phenomenon that has been investigated in HS patients. The data in this regard is conflicting. In one study including 50 HS patients and 50 healthy controls, bacterial DNA was more likely to be detected in the peripheral blood of HS patients. The DNA of Gram-negative bacteria, particularly *Escherichia coli,* appeared to predominate in the patient cohort [67]. The identification of bacterial DNA in HS patients was significantly correlated with increased levels of TNF-α, IL-1β, and Il-17 [67]. Conflicting results were obtained in a smaller study comparing 27 HS patients and 26 healthy controls using 16S rRNA sequencing. In the latter study, no significant differences were established between the bacterial composition of patient and control groups [66]. In such studies, possible contamination from skin-borne bacteria during transcutaneous needling is an important caveat that restricts the interpretation of these findings [66]. Ablative surgical treatments for HS, such as CO_2_ laser, do not appear to increase the relative risk of bacteraemia in patients [115].

Further serological evidence supporting the role of microbiome in inflammatory disease is the presence of anti-*Saccharomyces cerevisiae* antibodies (ASCA) in HS patients, which have also been associated with body fat mass [116,117]. It has been suggested that ASCA positivity may be associated with a specific underlying HS genotype as this antibody has been associated with Crohn’s disease patients harbouring specific *NOD2* variants [118,119]. 

### 3.8. Targeting the Microbiome as a Treatment Modality 

#### 3.8.1. Antibiotics 

Targeting the microbiome appears to be a logical option in the management of HS patients. Characterizing skin microbiome with the use of genomic techniques can have direct bench-to-bedside applications. This was exemplified in a patient suffering from comorbid recalcitrant acne fulminans and HS, who was successfully treated with targeted antibiotics after lesional skin microbiome was elucidated using NGS [120]. Antibiotic agents have also been successfully used to induce HS-remission in patients that have syndromic forms of the disease [121].

Tunnels pose another challenge in the medical management of HS. Their significance is reflected in the Hurley Staging system (a frequently employed static severity score), as their presence distinguishes patients with moderate-to-severe HS from a mild form of the disease. Rifampicin has exclusively been shown to efficiently hinder planktonic growth and eliminate biofilm in tunnels of HS patients [90,93]. Promising results have been obtained after sinus tracts were injected with triamcinolone. However, the definitive treatment of sinus tracts remains to be surgical excision [122].

Despite the importance of antibiotics for the treatment of patients suffering from HS, antibiotic resistance in HS patients is significant and on the rise. The prevalence of antibiotic resistance may be as high as 84.7% for tetracyclines, 69.3% for rifampicin, and 65.7% for clindamycin [123]. The former are antibiotics which are recommended for the treatment of HS by European, British, and American guidelines [92]. Apart from the judicious use of antibiotics and the timely initiation of biologic agents to prevent recurrent flares, other measures can be adopted to reduce antibacterial resistance. For example, antimicrobial washes may alter antibiotic resistance in HS lesions, particularly that of cephalosporins [124].

It is relevant to consider that antibiotics, including tetracyclines (considered first-line treatment for HS), could be contributing to disease remission not only by their bacteriostatic or bactericidal effects, but also through their additional anti-inflammatory and anti-collagenolytic properties [125]. Such observations highlight the underlying inflammatory nature of HS, and the increased risk of other co-morbid metabolic disease (including metabolic syndrome) characterized by a chronic, low-grade inflammatory response [126]. The identification of HS molecular mediators may allow for drug repurposing through the elucidation of the drug–gene interaction, paving the way for novel targeted therapeutic agents [127]. 

#### 3.8.2. Enteric Microbiome and Diet 

Beyond the use of antibiotics, the modification of the enteric microbiome has the propensity to alter disease trajectory. The use of probiotics as a treatment for HS has been suggested as a potential treatment option by restoring the cutaneous microbiome, replenishing *Cutibacterium* spp., *Corynebacterium,* and *Staphylococcus* and thereby restoring a healthy microbiome, which is less abundant in Gram-negative anaerobes [68,128]. 

Post-operative dietary restriction of brewer’s yeast in patients with specific anti-*Saccharomyces cerevisiae* IgGs has also been successful in inducing disease remission. Intriguingly, patients experience disease relapse upon re-exposure to the single-cell fungus [129]. The exclusion of dietary *Saccharomyces cerevisiae* over a longer period also promoted disease remission in HS patients [130]. 

As previously described, obesity and HS are strongly linked, with obese HS patients having a distinct HS phenotype [131]. Mice grown on a high-fat diet have been shown to have an increased local production of inflammatory cytokines such as TNF-α, IL-1β, and IL-6. These cytokines were not found to be elevated in the same mice at week 8 on a high-fat diet. However, levels of serum TNF-α and IFN-γ were elevated at week 12 and were found to positively correlate with weight [112]. This suggests that diet may alter the gut microbiome and local inflammatory cytokines, whilst obesity, which is intricately linked to diet, may contribute to more systemic forms of inflammation. 

### 3.9. The Microbiome—A Potential HS Biomarker?

Biomarkers are defined as “objective indications of medical state observed from outside the patient, and which can be measured accurately and reproducibly” [132]. Biomarkers can help predict, diagnose, and prognosticate patients. They have been valuable tools in the management of other inflammatory skin conditions such as psoriasis. However, they have not yet been extensively characterized and validated in HS [133]. The skin microbiome has the potential of acting as a biomarker [134]. 

#### 3.9.1. Prognostic Biomarker

The microbiome has the capacity to function as a prognostic biomarker in HS. Patients that have severe (Hurley stage III) disease harbor a greater number of bacterial species [91]. Isolating bacteria such as *Streptococci*, *Enterobacteriaceae,* and obligate anaerobic Gram-negative rods has also been associated with more severe disease [91]. Similarly, moderate but significant positive correlations have been established between HS disease severity and the relative abundance of mixed anaerobes. Moderate negative correlations have been made for major skin commensals, including *Cutibacterium* and *Corynebacterium spp.* [69]. Conflicting results have been obtained with regard to the correlation of HS severity and *S. aureus* [69,91,135].

ASCA positivity has also been associated with severe (Hurley Stage 3) disease [116]. The presence of bacterial DNA in peripheral blood is associated with significantly higher levels of pro-inflammatory cytokines such as TNF-α, IL-1β, and IL-17, but does not correlate with disease severity [67].

#### 3.9.2. Risk Biomarker 

In HS patients, changes in the cutaneous microbiome can precede clinical evidence of the disease and can therefore theoretically act as a risk biomarker. Non-affected skin of HS patients has been shown to lack bacterial aggregates at the stratum corneum of the epidermis as well as in hair follicles, as shown by PNA-FISH and confocal laser scanning microscopy when compared to the skin of healthy controls [94].

#### 3.9.3. Pharmacodynamic Biomarker

The microbiome shows potential as a pharmacodynamic biomarker. In a small cohort study, patients with moderate-to-severe HS who carry *S. aureus* nasally were found to be more likely to experience HS flares that lead to the failure of adalimumab therapy at week 12 than those who do not carry the bacterium (OR 8.67, *p* = 0.014) [136].

## 4. Discussion

A complex relationship exists between the microbiome, inflammation, and HS (Figure 1). HS, as an inflammatory condition, is subject to the effects of dysbiosis, whether cutaneous or enteric. Various studies have investigated the relationship between the disease and the microbiome, using an array of sampling and detection methods. Apart from highlighting the yearning of the HS research community to fully understand the pathophysiology of HS, they also bring to the fore the importance of global harmonization in sampling methodology in HS research [137].

Beyond its role in pathophysiology, the microbiome can potentially be targeted as a treatment option for altering disease trajectory. It has been shown that most antibiotics used in the treatment of HS exert anti-inflammatory properties. However, research is lacking with regard to the alteration of the microbiome after antibiotic therapy or any therapeutic intervention [138]. The use of dietary modification as a therapeutic intervention in the management of disease has several advantages, including the lack of associated negative effects and a low financial burden on the patient. The care of patients with restrictive diets should be overseen by nutritionists, who are ideally familiar with HS and its management as part of the multidisciplinary team providing personalized care to HS patients. As outlined above, advances in genomic technologies can facilitate the bench-to-bedside transition by guiding the management of complex HS cases. Apart from its therapeutic roles, the microbiome can potentially serve as a disease biomarker for the facilitation of disease classification, patient stratification, and the monitoring of treatment response. However, further research is required for the identification and validation of such tools. Furthermore, the microbiome can also help refine genotype-phenotype associations, this being another lacuna in our understanding of HS [139]. 

The studies outlined in this review provide compelling evidence for the dysregulation of the lesional and peri-lesional skin microbiome in HS. However, a direct cause–effect relationship cannot be assumed, and several questions arise. Primarily, the role of non-lesional cutaneous microbiota in the pathogenesis of HS is not completely understood. More specifically, the characterization of the evolution of microbial composition and abundance during the transition from unaffected to lesional skin should provide insight into causative trajectories leading to HS. One can hypothesize on whether changes in microbiota antedate the development of HS, thus representing a latent or pre-clinical stage of the disease. Alternatively, it is possible that the observed cutaneous dysbiosis in HS partly reflects an adaptive microbial response to acute inflammation in active HS lesions. Physiologically, it is likely that the microbiota contributes to causality, progression, and the maintenance of HS against a complex background of host immune response. These factors jointly determine the ensuing clinical picture and response to therapy. 

In conclusion, HS is a complex condition, with changes preceding clinical signs and symptoms, and differing according to lesion and anatomical location. Possible future research can be directed towards investigating whether environmental exposure (the hygiene hypothesis) in childhood is related to dysbiosis and disease trajectory in later life. An elegant unifying pathophysiological explanation for HS has not yet been established. Unravelling the role of the microbiome in HS will bring us a step closer to this end.

## Figures and Tables

**Figure 1 vaccines-09-01076-f001:**
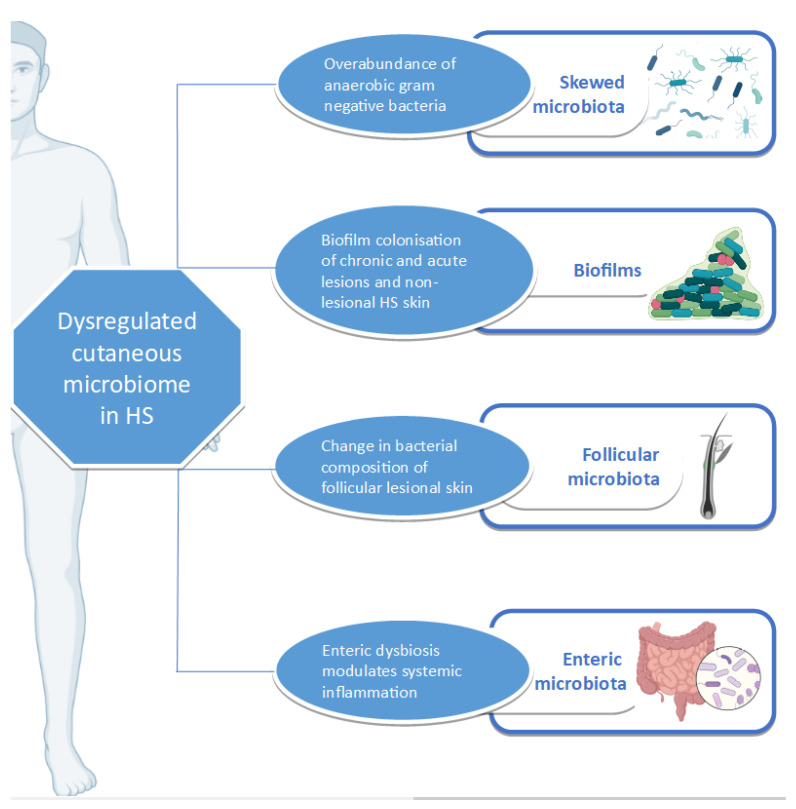
Events leading to cutaneous dysbiosis in HS patients.

**Table 1 vaccines-09-01076-t001:** Studies employing next generation sequencing targeting 16S rRNA to unravel the microbiome in various specimen types of HS patients.

Study Group	Specimen	Findings	Comments
Riverain-Gillet et al. [63]	Skin surface of non-lesional skin swab	Fourteen bacterial taxa statistically discriminated HS from healthy control skin were predominantly *Prevotella* but also *Actinomyces, Campyobacter ureolyticus,* and *Mobiluncus.*	Bacterial richness is similar between HS skin and skin of healthy controls. Evenness is significantly higher in HS skin samples. The microbiome of clinically unaffected intertriginous skin in HS patients is characterized by an anaerobic shift contrasting with a decreased abundance of aerobic skin commensals.
H C Ring et al. [64]	Lesional HS skin biopsy	Dominated by *Corynebacterium* spp., *Prophyromonas,* and Peptoniphilus spp.	No difference in richness between three groups.
	Non-lesional HS skin biopsy	Dominated by *Acinetobacter* and *Moraxella* spp.	
	Healthy control skin biopsy	Equally distributed microbiome.	
H C Ring et al. [65]	Tunnels	Dominated by anaerobic species (*Prophyromonas* spp *and Prevotella* spp.).	
H C Ring et al. [66]		Consistent levels of various bacteria such as *Acinetobacter* and *Moraxella,* which do not differ significantly between control and HS blood.	Source of bacteria is likely from skin of the antecubital fossa discovered due to high sensitivity of NGS.
Hispan et al. [67]		Presence of bacterial DNA was significantly higher in HS group vs. control. Dominant species was *E. coli.*	

## Data Availability

Not applicable.

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
