# Peer review of "The Clinical Relevance of the Microbiome in Hidradenitis Suppurativa: A Systematic Review"

_vaccines, 2021, doi:10.3390/vaccines9101076_

Round 1
Reviewer 1 Report
The paper is interesting, well-written and gives a comprehensive review of the newest knowledge on the topic.
Minor comments:
1. Differences in skin microbiota in HS lesions do not automatically confirm a causal effect. It may be that the changes in active HS lesions are the cause of the switch in microbiota. The microbiota in non-lesional skin may be of bigger importance than lesional microbiota. In the paper, you do not seem to discuss this.
2. 3.8.1. Antibiotics: Are there efficient antibiotics for all the species that are found and of relevance for developing HS? It is almost impossible to change one's microbiome with antibiotics, even if temporarily optimizing it. Do antibiotics even penetrate the tunnels and all cicatricial tissue? I think the claim on antibiotic's efficiency is a bit simplistic, having in mind all the previous explanations throughout the paper.
3. The nomenclature has changed and it is now Cutibacterium acnes, not Propionobacterium! In the beginning of the paper you use Propionobacterium, further on Cutibacterium. Make sure you use the correct nomenclature throughout.
Author Response
The paper is interesting, well-written and gives a comprehensive review of the newest knowledge on the topic.
Thank you for your positive feedback
Minor comments:
- Differences in skin microbiota in HS lesions do not automatically confirm a causal effect. It may be that the changes in active HS lesions are the cause of the switch in microbiota. The microbiota in non-lesional skin may be of bigger importance than lesional microbiota. In the paper, you do not seem to discuss this.
Thank you pointing this out. We have included the following paragraph in the discussion to highlight this observation.
“The studies outlined in this review provide compelling evidence for dysregulation of the lesional and peri-lesional skin microbiome in HS. However, a direct cause-effect relationship cannot be assumed, and several questions arise. Primarily, the role of non-lesional cutaneous microbiota in the pathogenesis of HS is incompletely understood. Specifically, characterisation of the evolution in microbial composition and abundance during the transition from unaffected to lesional skin, should provide insight into causative trajectories leading to HS. One can hypothesize on whether changes in microbiota antedate the development of HS, thus representing a latent or pre-clinical stage of the disease. Alternatively, it is possible that the observed cutaneous dysbiosis in HS partly reflects an adaptive microbial response to acute inflammation in active HS lesions. Physiologically, it is likely that microbiota contribute to causality, progression and maintenance of HS against a complex background of host immune response. These factors jointly determine the ensuing clinical picture and response to therapy. “
- 3.8.1. Antibiotics: Are there efficient antibiotics for all the species that are found and of relevance for developing HS?
Thank you for this observation. We have referred throughout the text regarding guideline-recommendations on the use of antibiotics for HS. However, we have included the following sentence to describe the other potential effects
It has been shown that most antibiotics used in the treatment of HS exert anti-inflammatory properties. However, research is lacking with regards to alteration of the microbiome after antibiotic therapy or any therapeutic intervention (138).
It is almost impossible to change one's microbiome with antibiotics, even if temporarily optimizing it. Do antibiotics even penetrate the tunnels and all cicatricial tissue?
Thank you for making this point. We have highlighted the difficulties with treating sinus tracts medically by including this sentence in section 3.8.1;
“Promising results have been obtained after sinus tracts were injected with triamcinolone, however, the definitive treatment of sinus tracts remains surgical excision (122).”
I think the claim on antibiotic's efficiency is a bit simplistic, having in mind all the previous explanations throughout the paper.
Thank you. We agree with this comment and have added this sentence in 3.8.1 to drive the point further.
It is relevant to consider that antibiotics, including tetracyclines (considered first line treatment for HS) could be contributing to disease remission not only by their bacteriostatic or bactericidal effects, but also through their additional anti-inflammatory and anti-collagenolytic properties (125). Such observations highlight the underlying inflammatory nature of HS. Potentially, they can explain the increased risk of co-morbid metabolic disease characterized by a chronic, low-grade inflammatory response (126). The identification of HS molecular mediators may allow for drug repurposing through elucidation of drug-gene interaction, paving the way for novel targeted therapeutic agents (127).
- The nomenclature has changed and it is now Cutibacterium acnes, not Propionobacterium! In the beginning of the paper you use Propionobacterium, further on Cutibacterium. Make sure you use the correct nomenclature throughout.
Thank you for highlighting this. We updated the manuscript to reflect the change in nomenclature.
Reviewer 2 Report
This is a great review paper on Hidradenitis suppurativa.
I have a few comments:
- Propionibacterium acnes has been re-named as Cutabacterium acnes. All should be corrected into C. acnes.
- Despite of being called "acne inversa" and having similar treatment regimens, HS does not seem to have predominance of C. acnes. What would be the reason for this difference?
- Are there any articles that have examined the changes in the skin microbiome before and after treatment (i.e. isotretinoin or antibiotics?)
Author Response
This is a great review paper on Hidradenitis suppurativa.
Thank you for your positive feedback.
I have a few comments:
1. Propionibacterium acnes has been re-named as Cutabacterium acnes. All should be corrected into C. acnes.
Thank you for pointing this out. We have updated the manuscript to reflect the change in nomenclature.
2. Despite of being called "acne inversa" and having similar treatment regimens, HS does not seem to have predominance of C. acnes. What would be the reason for this difference?
Thank you for making this interesting observation. Despite having similar pathophysiological processes (principally occlusion of the infundibular portion of the PSU), one of the main differences is that acne involves an oily microenvironment which would favor a different microbiome. We have revised the manuscript and made reference to this fact in the introduction which now reads:
The name Hidradenitis Suppurativa is misleading as it suggests the condition to be the result of purely infective processes involving the apocrine glands. This name is the subject of controversy, with some recommending that the disease should be renamed Acne Inversa(3,4). The latter name conceptualizes the fact that unlike acne, patients with HS experience deep seated abscesses which involve skin that is not usually affected by acne (intertriginous skin), and which therefore hosts a distinct microbiome (5).
3. Are there any articles that have examined the changes in the skin microbiome before and after treatment (i.e. isotretinoin or antibiotics?)
Thank you for raising this issue. Research is lacking in this area. We have updated the text to reflect this by including this comment in the discussion;
“It has been shown that most antibiotics used in the treatment of HS exert anti-inflammatory properties. However, research is lacking with regards to alteration of the microbiome after antibiotic therapy or any therapeutic intervention (138)”